# One-Stage Femtosecond Laser-Assisted Deposition of Gold Micropatterns on Dielectric Substrate

**DOI:** 10.3390/ma15196867

**Published:** 2022-10-03

**Authors:** Tatiana Lipateva, Alexey Lipatiev, Sergey Lotarev, Georgiy Shakhgildyan, Sergey Fedotov, Vladimir Sigaev

**Affiliations:** Department of Glass and Glass-Ceramics, Mendeleev University of Chemical Technology, 9 Miusskaya Sq., 125047 Moscow, Russia

**Keywords:** gold nanoparticles, selective metallization, gold electrodes, direct laser writing, laser-induced backside wet etching, chemical liquid-phase deposition

## Abstract

In this study, a simple one-stage laser-assisted metallization technique based on laser-induced backside wet etching and laser-induced chemical liquid-phase deposition is proposed. It allows for the fabrication of gold micropatterns inside the laser-written trace on a glass substrate. The reduction and deposition of gold inside and outside the laser-ablated channel were confirmed. The presence of Au nanoparticles on the surface of the laser-written micropattern is revealed by atomic force microscopy. The specific resistivity of the gold trace formed by ultrafast light-assisted metal micropatterning on a dielectric glass substrate is estimated as 0.04 ± 0.02 mΩ·cm. The obtained results empower the method of the selective laser-assisted deposition of metals on dielectrics and are of interest for the development of microelectronic components and catalysts, heaters, and sensors for lab-on-a-chip devices.

## 1. Introduction

The miniaturization of optical sensors and the rapid development of microelectronics, microfluidics, and lab-on-a-chip technologies cause an urgent demand for effective methods of the spatially selective deposition of metals on the glass substrate. Though different methods, such as electroplating, electroless plating [1], and photolithography [2], are widely used for the metallization of dielectrics, the application of direct laser writing, which is a maskless and versatile technique for the precise 3D microstructuring of various types of materials, generates considerable interest in this field [3]. Over the past decade, approaches to the laser-assisted metallization of dielectrics based on laser-induced chemical liquid phase deposition have been well developed [4]. Most studies were performed using continuous-wave [5] or nanosecond [6,7] laser sources for copper deposition while metal salt solution served as a light absorber. However, femtosecond laser irradiation provides the possibility of using any solutions, regardless of their absorption, better precision in the formation of complex metal structures, and control of their morphology while maintaining high process efficiency [8,9].

As gold has excellent chemical stability and physical properties, the direct laser writing of gold micropatterns is essential for the fabrication of microelectronic components, microheaters [10], molecular sensors [11], and surface-enhanced Raman scattering (SERS) substrates [12]. On the other hand, to date, the investigations of femtosecond laser-assisted gold deposition on the dielectric substrate are very limited and are mainly represented by multistage and time-consuming procedures [13,14] or include the use of complex Au-containing photoresists or solutions which enable the multiphoton photoreduction [15]. Moreover, until now, the gold micropatterning was shown only as the surface gold deposition. However, it is known that such metal surface structures are not wear-resistant and can be easily removed under external action [6]. The aim of our work is to develop a simple one-step method for the robust selective metallization of dielectric using a solution without any additives.

In this study, for the first time, a one-stage technique of laser-assisted selective gold deposition on glass is proposed and features of gold micropatterns directly laser-written inside the laser-written trace on glass substrate are investigated. The simplicity and robustness of the proposed technique are provided by ultrashort laser pulses inducing the formation of ablated channels inside and on the walls of which gold is deposited from HAuCl_4_ solution without any photosensitive additives.

## 2. Materials and Methods

Menzel Gläser microscope slides (Thermo Scientific, Braunschweig, Germany) fabricated from soda-lime glass were used as a substrate for laser-assisted gold deposition experiments. Glass samples were prepared by cutting into 10 mm × 5 mm × 1 mm plane-parallel plates. A FemtoLab laser micromachining setup based on Pharos SP femtosecond Yb:KGW laser (Light Conversion Ltd., Vilnius, Lithuania) emitting 1030 nm pulses with 180 fs duration was used for direct laser writing. The pulse energy was tuned in the range of 100 to 800 nJ, and the repetition rate was set to 500 kHz to provide heat accumulation and the thermal mode of laser processing which is favorable for metal reduction and deposition [5]. The scheme of laser-assisted gold micropatterning is shown in Figure 1. 

The glass sample was placed in a silica cuvette and fully immersed in 4 M HAuCl_4_ aqueous solution. Translation of the cuvette with the sample was performed using an Aerotech ABL1000 air-bearing 3D motorized stage synchronized with laser pulse generation. The linear polarization of the laser beam was aligned along the scanning direction. We used an inverted setup where the laser beam was focused through the bottom of the cuvette onto the upper surface of the sample (Figure 1a). This configuration allows excluding the negative influence of the cavitation bubbles on the focusing conditions and gold deposition process because gravitational forces ensure that the bubbles easily leave the laser writing region. A two-hole spatial filter with φ = 130° and θ = 23° was introduced into the optical scheme in front of the objective lens Olympus LCPLNIR 20X (N.A. = 0.45) for the aberration-free laser beam focusing at a large depth [16].

Reflected light microscopy of laser-written gold micropatterns was carried out by an Olympus BX51 optical microscope equipped with an Olympus DP73 CCD camera. The nanoscale structure of the gold micropatterns was investigated using atomic force microscopy (AFM) in an NTegra Spectra (NT-MDT Ltd., Zelenograd, Russia) nanolaboratory in semi-contact mode. The X-ray diffraction pattern of the 1 × 1 mm^2^ gold square was recorded on a Bruker D2 PHASER diffractometer (CuKα radiation, Ni filter) in the angular range of 35–80° (step size 0.02°). Electrical measurements on laser-written long gold traces connected with square gold pads at their ends (Figure 1b) were performed using a Mastech MY62 multimeter.

## 3. Results and Discussion

The ablation of the surface of the glass sample and the simultaneous appearance of cavitation bubbles indicated the focusing of the laser beam at the glass–solution interface. After adjusting the focus, the direct laser writing of gold micropatterns was carried out. The reflected light microscope images of laser-written tracks are shown in Figure 2a–c, which highlight a distinct gold tint of light reflected from the tracks. The X-ray diffraction pattern registered for the 1 mm × 1 mm square hatched with a laser beam at a pulse energy of 500 nJ and translation speed of 1 mm/s with a step of 10 µm confirms the direct laser-induced deposition of crystalline gold (Figure 2d). 

Generally, a few passes of the laser beam are required for the fabrication of continuous gold traces. The first pass of the laser beam at 0.1 mm/s speed produced the ablated groove with rough walls on which large gold aggregates were randomly deposited. The rough surface of the laser-ablated groove is favorable for both adhesion of metal to glass and the formation of metal seeds [6]. Moreover, the laser-induced ablation of the material surface in contact with metal likely activates the chemical reaction and produces the seeding layer containing not only metal monomers but also corresponding metal compounds [17,18]. In our case, HAuCl_4_ solution can react with multicomponent glass under extremal laser-induced conditions, forming a chemically bonded layer, which is also favorable for strong metal-to-glass adhesion. Nevertheless, further study is in progress to better understand the adhesion nature of laser-deposited gold micropatterns. 

The second and subsequent passes enable intensive gold deposition so that gold is formed not only in the groove but also outside it on the glass surface. It should be noted that opposite to copper deposition, where a reducing agent in the solution and alkaline condition is required to launch a laser-induced chemical reaction of copper reduction [5], exposure to femtosecond pulses opens a way to the direct laser-induced reduction of metal due to the availability of free electrons produced via multiphoton ionization. Optical images of the laser-written tracks show that even a single laser beam pass is sufficient for gold reduction and deposition. We consider that these processes are also facilitated by the heat accumulation effect provided by a high pulse repetition rate. Thus, the application of ultrafast laser pulses focused on the backside surface of a sample immersed in a metal salt solution realizes combining laser-induced backside wet etching and laser-induced chemical liquid-phase deposition processes in one stage as it allows producing the groove, the seeding layer on its walls, and deposited gold. The cross-section view (Figure 2b) of laser-written gold micropatterns is generally represented by two parts. The first part is located in the ablated groove, with a depth of up to 10 μm, while the second one, 1–3 μm thick, lies on the glass surface. The latter has a weaker adhesion to glass and can be torn off mechanically. It is confirmed by ultrasonication in water for 1 min (Figure 2c). At one or two laser beam passes, increasing the laser pulse energy leads to the expansion of the width and depth of the ablated channel as well as the metalized region (Figure 2a). However, when applying a large number of passes, the width of the upper part of the gold traces tends to a practically constant value of ~40 μm, which significantly exceeds the laser beam waist diameter of ~3.5 µm. This indicates the onset of equilibrium between the absorbed energy of the laser pulses, the reflection and scattering of radiation, and heat dissipation by the deposited gold. The size of the durable gold part located inside the groove on the glass surface can be controlled by changing the focusing parameters and the pulse energy used. A decrease in the pulse energy from 800 nJ to 500 nJ reduced the width of this part from 20 to 5 μm. Tighter laser beam focusing can be realized using an objective lens with a higher numerical aperture, which will open a way to reduce the width of the gold micropattern down to 1 μm.

The results of the morphology analysis of a laser-written gold micropattern are shown in Figure 3a,c. 

The structure of the gold micropattern is formed by aggregated gold nanoparticles (Au NPs). The size of the Au NPs evidently increases from the edge to the center, which can be associated with the Gaussian profile of the beam, i.e., more intense ionization and heating occur in the center. Moreover, a strong electric field and pressure in the area close to the focused laser beam waist induce elongation of the Au NPs, their sintering, and denser packing.

The current–voltage curve (Figure 3b) was measured on three 2 mm long gold micropatterns with 1.8 ± 0.9∙10^−4^ mm^2^ cross-section areas measured in different crosscuts of the traces. The average resistance of this conductive structure equal to 4.68 Ω was deduced from this curve. The specific resistance of the deposited gold in the laser-written trace was estimated as 0.04 ± 0.02 mΩ·cm. The obtained value is rather high in comparison, with the resistivity of bulk gold (0.0023 mΩ·cm) due to the presence of pores formed during the gold nanoparticles aggregation. However, it is comparable with the results demonstrated for laser-deposited copper electrodes [5,6]. This indicates satisfactory quality and potential in electrical applications of laser-fabricated gold traces.

## 4. Conclusions

One-stage femtosecond laser-assisted fabrication of gold micropatterns on glass substrate was demonstrated for the first time. The microstructure of laser-written gold traces is shown to be represented by compacted nanoparticles whose packing is denser in the center of the trace than near its edges. The electric resistivity of gold in the traces was estimated as ~0.04 ± 0.02 mΩ·cm. In future work, it is planned to investigate the nature of gold-to-glass adhesion and optimize the concentration of the metal salt solution in order to fabricate gold micropatterns with reduced porosity and improved electrical properties. Moreover, a tighter laser beam focus or elongated laser beam waist along the scanning direction will help to control the morphology of the deposited gold and reduce the size of the grooves filled with gold, which is important for microelectronic applications. The obtained results pave the way for novel microfabrication techniques for the development of microelectronic components and catalysts, heaters, and sensors for lab-on-a-chip devices.

## Figures and Tables

**Figure 1 materials-15-06867-f001:**
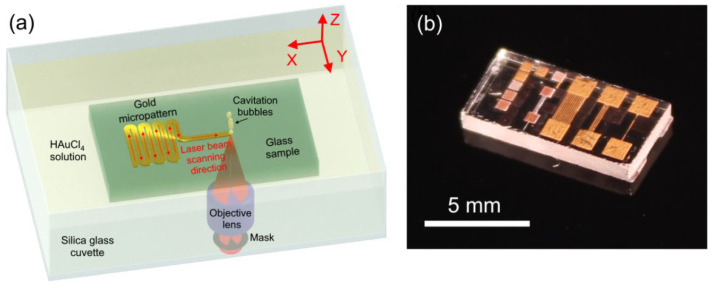
(**a**) Scheme of one-step laser-assisted gold micropatterning on glass substrate. (**b**) Photograph of glass sample with deposited gold micropatterns.

**Figure 2 materials-15-06867-f002:**
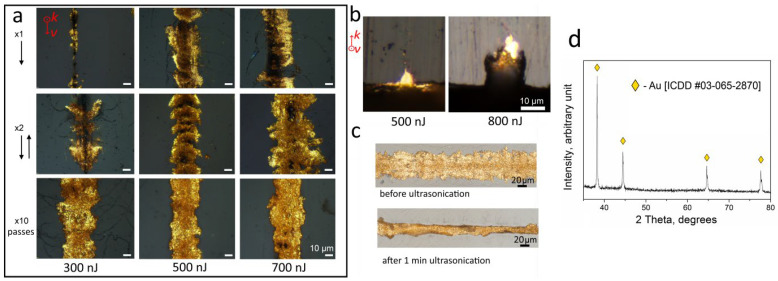
(**a**) Reflected light microscope images of the laser-deposited gold traces at different pulse energies and numbers of laser beam passes; (**b**) Reflected light microscope images of the cross-section of gold traces fabricated by 100 laser beam passes; (**c**) laser-deposited gold traces before and after ultrasound treatment (**d**) X-ray diffraction pattern of the laser-written Au square. Vectors v and k indicate laser writing direction and laser beam propagation direction, respectively.

**Figure 3 materials-15-06867-f003:**
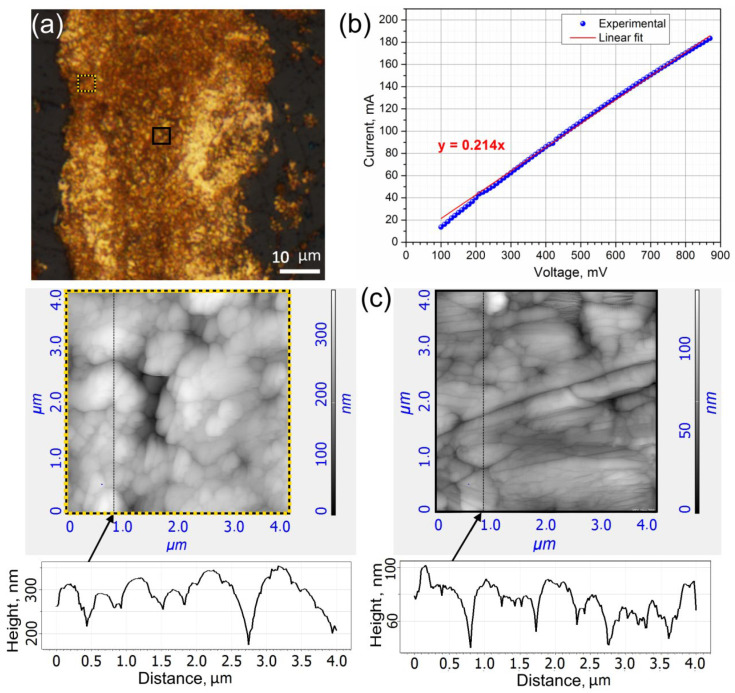
(**a**) Brightfield optical image of gold trace deposited at 500 kHz, 500 nJ, 1 mm/s by three laser beam passes corresponding to 4500 pulses per irradiated spot; (**b**). Current–voltage curve for the laser-deposited gold micropatterns; (**c**) AFM images and corresponding profiles registered for the gold track. Regions of interest are indicated by yellow dotted and black solid squares in (**a**).

## Data Availability

Not applicable.

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
