# Peer review of "One-Stage Femtosecond Laser-Assisted Deposition of Gold Micropatterns on Dielectric Substrate"

_materials, 2022, doi:10.3390/ma15196867_

Round 1

Reviewer 1 Report

In this work, the authors employed a femtosecond laser to produce selective gold deposition on glass. The morphologies and the specific resistivity of the deposited gold were measured. However, some points of this research should be clarified. Therefore, a minor revision is required to be qualified to publish in Materials. In detail: 

1)     Please add a description for k, v in figure 2.

2)     Please explain more for the statement “So the application of ultrafast laser pulses realizes combining laser-induced backside wet etching and laser-induced chemical liquid-phase deposition processes in one stage” in lines 114 and 115.

3)     What is the thickness of the deposited gold?

4)     How can the femtosecond laser processing create the gold deposition on glass immersed in HauCl4?

5)     What is the bonding strength of the deposited gold?

6)     What is the best pulse energy in this study for the gold deposition?

7)     In figure 3, the morphology of the deposited gold in the solid black squares seems to differ from that in the yellow dotted squares. Please clarify this point.

Author Response

Dear Reviewer,

We thank you for your careful reading of our article. We took great care to respond to each remark or comment that will help us to improve the quality of our manuscript. A point-by-point response is included below and all modifications are tracked in the revised manuscript.

Reviewer #1

In this work, the authors employed a femtosecond laser to produce selective gold deposition on glass. The morphologies and the specific resistivity of the deposited gold were measured. However, some points of this research should be clarified. Therefore, a minor revision is required to be qualified to publish in Materials. In detail:

1)     Please add a description for k, v in figure 2.

We have added the following description in figure 2 “Vectors v and k indicate laser writing direction and laser beam propagation direction respectively.”

2)     Please explain more for the statement “So the application of ultrafast laser pulses realizes combining laser-induced backside wet etching and laser-induced chemical liquid-phase deposition processes in one stage” in lines 114 and 115.

Thank you for pointing this out. We have expanded the statement to make it clearer: “So the application of ultrafast laser pulses focused on the backside surface of a sample immersed in a metal salt solution realizes combining laser-induced backside wet etching and laser-induced chemical liquid-phase deposition processes in one stage as it allows producing the groove, the seeding layer on its walls and deposited gold.”

3)     What is the thickness of the deposited gold?

A cross-section view (Fig. 2(b)) of the laser-written gold micropattern is generally represented by two parts. The first part is located in the ablated groove with a depth of up to 10 μm, while the second one, 1-3 μm thick, lies on the glass surface. The size of the durable gold part located inside the groove on glass surface can be controlled by changing the focusing parameters and the pulse energy used. A decrease of the pulse energy from 800 nJ to 500 nJ reduced the width of this part from 20 to 5 μm. Tighter laser beam focusing can be realized using an objective lens with a higher numerical aperture which will open a way to reduce the width of the gold micropattern down to 1 μm.

4)     How can the femtosecond laser processing create the gold deposition on glass immersed in HauCl4?

We describe the process of the femtosecond laser-induced gold deposition on glass immersed in HauCl4 as follows: “The first pass of the laser beam at 0,1 mm/s speed produced the ablated groove with rough walls on which gold agglomerations were randomly deposited. The rough surface of the laser-ablated groove is favorable for both adhesion of metal to glass and the formation of metal seeds [Seo, J.M.; Kwon, K.K.; Song, K.Y.; Chu, C.N.; Ahn, S.H. Deposition of Durable Micro Copper Patterns into Glass by Combining Laser-Induced Backside Wet Etching and Laser-Induced Chemical Liquid Phase Deposition Methods. Materials (Basel). 2020, 13, doi:10.3390/ma13132977.]. Moreover, laser-induced ablation of material surface in contact with metal likely activates the chemical reaction and produces the seeding layer containing not only metal monomers but also corresponding metal compounds [Huang, Yajun, et al. "Robust metallic micropatterns fabricated on quartz glass surfaces by femtosecond laser-induced selective metallization." Optics Express 30.11 (2022): 19544-19556; Broadhead, Eric J., and Katharine Moore Tibbetts. "Fabrication of Gold–Silicon Nanostructured Surfaces with Reactive Laser Ablation in Liquid." Langmuir 36.34 (2020): 10120-10129.]. In our case HAuCl4 solution can react with multicomponent glass under extremal laser-induced conditions forming a chemically bonded layer which is also favorable for strong metal-to-glass adhesion. Nevertheless, further study is in progress to better un-derstand the adhesion nature of laser-deposited gold micropattern. The second and subsequent passes enable intensive gold deposition so that gold is formed not only in the groove but also outside it on the glass surface. It should be noted that opposite to copper deposition, where a reducing agent in the solution and alkaline condition are required to launch a laser-induced chemical reaction of copper reduction [Kim, H.G.; Park, M.S. Fast Fabrication of Conductive Copper Structure on Glass Material Using Laser-Induced Chemical Liquid Phase Deposition. Appl. Sci. 2021, 11, doi:10.3390/app11188695.], exposure to femtosecond pulses opens a way to the direct laser-induced reduction of metal due to the availability of free electrons produced via multiphoton ionization.”

5)     What is the bonding strength of the deposited gold?

We appreciate this useful question very much. The laser-written gold micropattern consists of two parts: the first part is located in the ablated groove and represented by aggregated gold nanoparticles that adhere to the rough surface of the glass inside the groove. This part of the micropattern endured ultrasonication in water for 1 minute. The second part, 1-3 μm thick one, lies on the glass surface and consists of gold nanoparticles and aggregates that cover a polished smooth glass surface so they can be easily removed by scribing with soft abrasives or during ultrasonication in water. The study of metal-to-glass adhesion nature is in progress now.

6)     What is the best pulse energy in this study for the gold deposition?

The best pulse energy for laser-induced deposition of gold is 500 nJ at a pulse repetition rate of 500 kHz (taking into account the pulse energy step of 200 nJ), scanning speed of 1 mm/s and 3 laser beam passes.

7)     In figure 3, the morphology of the deposited gold in the solid black squares seems to differ from that in the yellow dotted squares. Please clarify this point.

The size of the Au NPs evidently increases from the edge to the center which can be associated with the Gaussian profile of the beam, i.e. more intense ionization and heating occur in the center. Moreover, a strong electric field and pressure in the area closer to the center of the focused laser beam waist induce elongation of the Au NPs, their sintering and more dense packing.

Reviewer 2 Report

This manuscript has great innovative significance in investigating one-stage femtosecond laser-assisted deposition of gold 2 micropatterns on dielectric substrate. The work can arouse wide interests of researchers in design and preparation of new functional materials. The manuscript is interesting. In my frank opinion, the manuscript should be deserved for its final publication in such high-level Journal. The main reasons are as follows:

1. At first, the English ABSTRACT should be revised, and a unified simple present tense should be used.

2. The research significance and future work should be described in the final stage of the abstract.

3. Aims need to be concisely stated and added at the end of introduction. Not only what was done/investigated, but why.

4. Authors have mentioned laser parameters in section 2. Please provide the reason for selecting such data? OR refer to the literature that has been reported?

5. Under normal conditions, in conclusion section, important conclusions should be elaborated point by point for brevity and prominence, such as a) … … b) … … c) … ….

6. And also in the last point future research work should be given in conclusion section.

Author Response

Dear Reviewer,

We thank you for your careful reading of our article. We took great care to respond to each remark or comment that will help us to improve the quality of our manuscript. A point-by-point response is included below and all modifications are tracked in the revised manuscript.

Reviewer #2

This manuscript has great innovative significance in investigating one-stage femtosecond laser-assisted deposition of gold 2 micropatterns on dielectric substrate. The work can arouse wide interests of researchers in design and preparation of new functional materials. The manuscript is interesting. In my frank opinion, the manuscript should be deserved for its final publication in such high-level Journal. The main reasons are as follows:

Thank you very much for the appreciation of our work. We have revised the manuscript thoroughly to meet the requirements for publication in such high-level Journal.

  1. At first, the English ABSTRACT should be revised, and a unified simple present tense should be used.

Thanks for the useful comment. We have tried our best to revise the English in the Abstract section of the manuscript.

  1. The research significance and future work should be described in the final stage of the abstract.

We have expanded the Abstract section with research significance: “The obtained results empower the method of selective laser-assisted deposition of metals on dielectrics and are of interest for the development of microelectronic components and catalysts, heaters, and sensors for lab-on-a-chip devices.” Also, we have added the description of future work in Conclusions: “In future work, it is planned to investigate the nature of gold-to-glass adhesion and optimize the concentration of the metal salt solution in order to fabricate gold micropatterns with reduced porosity and improved electrical properties. Moreover, a tighter laser beam focusing or elongation laser beam waist along the scanning direction will help to control the morphology of deposited gold and reduce the size of the grooves filled with gold, which is important for microelectronic applications.” 

  1. Aims need to be concisely stated and added at the end of introduction. Not only what was done/investigated, but why.

By following your constructive suggestion, we have concisely stated the aim at the end of the Introduction:  “The aim of our work is to develop a simple one-step method for robust selective metallization of dielectric using a solution without any additives.”

  1. Authors have mentioned laser parameters in section 2. Please provide the reason for selecting such data? OR refer to the literature that has been reported?

The main reason for using selected laser parameters is to provide not only nonlinear absorption and plasma formation but also a heat accumulation and thermal mode of laser processing which is favorable for gold deposition [Kim, Han-Guel, and Min-Soo Park. "Fast Fabrication of Conductive Copper Structure on Glass Material Using Laser-Induced Chemical Liquid Phase Deposition." Applied Sciences 11.18 (2021): 8695].

  1. Under normal conditions, in conclusion section, important conclusions should be elaborated point by point for brevity and prominence, such as a) … … b) … … c) … ….

There are no mandatory requirements for the Conclusion section in Materials. Thus, we used standard practice to summarize the results of the study and future prospects in it.

  1. And also in the last point future research work should be given in conclusion section.

We have added our considerations for future research work in conclusions: “In future work, it is planned to investigate the nature of gold-to-glass adhesion and optimize the concentration of the metal salt solution in order to fabricate gold micropatterns with reduced porosity and improved electrical properties. Moreover, a tighter laser beam focusing or elongation laser beam waist along the scanning direction will help to control the morphology of deposited gold and reduce the size of the grooves filled with gold, which is important for microelectronic applications.”

Reviewer 3 Report

The manuscript entitled "One-Stage Femtosecond Laser-Assisted Deposition of Gold Micropatterns on Dielectric Substrate" and written by Tatiana Lipateva and co-workers presents a methodology to fabricate gold micropatterns over glass substrates by irradiating the substrate immersed in a HAuCl4 aqueous solution.

Even though the manuscript is well written in terms of the communication format, some information is still missing to view the current results' potential implications fully.

- Besides the mechanical removal of gold and 1 min ultrasonication, no additional discussion is provided about the Au adhesion over the glass substrate, i.e., there is no attempt to differentiate whether the Au is simply physisorbed or chemisorbed on the substrate. A clear differentiation could be done by analyzing the samples through XPS.

- Please replace the informal expression "pass" with the exact number of pulses delivered per irradiated spot. This will help the authors understand the scalability potential of their technique and the real implications for microelectronics, microfluidics, and lab-on-a-chip technologies.

-Even though the provided description of how the metal salt is reduced is not wrong, it is still incomplete. Please take a look at the following manuscript to complete it: https://doi.org/10.1021/acs.langmuir.0c01581

Author Response

Dear Reviewer,

We thank you for your careful reading of our article. We took great care to respond to each remark or comment that will help us to improve the quality of our manuscript. A point-by-point response is included below and all modifications are tracked in the revised manuscript.

Reviewer #3

The manuscript entitled "One-Stage Femtosecond Laser-Assisted Deposition of Gold Micropatterns on Dielectric Substrate" and written by Tatiana Lipateva and co-workers presents a methodology to fabricate gold micropatterns over glass substrates by irradiating the substrate immersed in a HAuCl4 aqueous solution.

Even though the manuscript is well written in terms of the communication format, some information is still missing to view the current results' potential implications fully.

Thanks for your positive evaluation of the manuscript. We are sorry for that some information was missing. We have responded to all comments carefully and made corresponding revisions in the manuscript.

  1. Besides the mechanical removal of gold and 1 min ultrasonication, no additional discussion is provided about the Au adhesion over the glass substrate, i.e., there is no attempt to differentiate whether the Au is simply physisorbed or chemisorbed on the substrate. A clear differentiation could be done by analyzing the samples through XPS.

We agree that X-ray photoelectron spectroscopy (XPS) is a powerful tool for the analysis of the chemical  bonding. The spectra obtained by this method indicate changes in the bond energy, which can be caused by the formation of new bonds or scattering from a rough surface or the shift in electron density from the surface layer to the matrix and charge transfer. The interpretation of the spectra is especially ambiguous for the case of a non-continuous sparse coating on the multicomponent substrate since the contribution of the substrate will be superimposed on the spectral data. This will complicate unambiguously interpreting the shift Au4f as forming a new bond, for example for ultrasmall clusters and single Au(0) the binding energy is 84.5-85 eV [DiCenzo, S. B., Berry, S. D., & Hartford Jr, E. H. (1988). Photoelectron spectroscopy of single-size Au clusters collected on a substrate. Physical Review B, 38(12), 8465] while metallic Au(0) 4f = 84 eV. So the task is not trivial and deserves a separate thorough and systematic study, including a number of samples preparation. Such study is in progress now and is out of the format of the communication paper. Nevertheless, we have added some discussion about the gold adhesion to glass substrate based on recent results of Huang et al., which showed that laser-induced etching (ablation) of glass in contact with metal activates chemical reaction and produces the layer containing not only metal monomers but also corresponding metal compounds [Huang, Yajun, et al. "Robust metallic micropatterns fabricated on quartz glass surfaces by femtosecond laser-induced selective metallization." Optics Express 30.11 (2022): 19544-19556]. So we believe that there is a chemically deposited gold layer on the walls of the ablated groove on the glass surface which forms under conditions of ionizing plasma generation, gold reduction and glass melting and evaporation.

  1. Please replace the informal expression "pass" with the exact number of pulses delivered per irradiated spot. This will help the authors understand the scalability potential of their technique and the real implications for microelectronics, microfluidics, and lab-on-a-chip technologies.

Thank you for the remark which helps to improve the understanding of the manuscript. We have provided the exact number of laser pulses delivered per irradiated spot for the best laser processing mode found in this study (please, see Fig. 3), i.e. 3 laser beam scanning passes at scanning speed of 1 mm/s, pulse repetition rate of 500 kHz correspond to 4500 laser pulses per irradiated spot, which has a diameter of about 3 µm.

  1. Even though the provided description of how the metal salt is reduced is not wrong, it is still incomplete. Please take a look at the following manuscript to complete it: https://doi.org/10.1021/acs.langmuir.0c01581

We are so grateful for the suggested reference which improves the completeness of our manuscript. This work and recent results obtained by Huang et al. clearly indicate the formation under reactive laser ablation the metal-silicon or metal-glass chemically bonded layer with corresponding positive valence metal substances which favorably serves for the adhesion increase and subsequent effective deposition of metal. Using these references we have expanded the discussion section accordingly.

Round 2

Reviewer 3 Report

The authors have adequately addressed the comments. Therefore, I can recommend to accept the current manuscript for its publication.